# Identification, Analysis and Gene Cloning of the SWEET Gene Family Provide Insights into Sugar Transport in Pomegranate (*Punica granatum*)

**DOI:** 10.3390/ijms23052471

**Published:** 2022-02-23

**Authors:** Xinhui Zhang, Sha Wang, Yuan Ren, Chengyan Gan, Bianbian Li, Yaoyuwei Fan, Xueqing Zhao, Zhaohe Yuan

**Affiliations:** 1Co-Innovation Center for Sustainable Forestry in Southern China, Nanjing Forestry University, Nanjing 210037, China; zhxinhui@njfu.edu.cn (X.Z.); wangsha@njfu.edu.cn (S.W.); renyuan426@njfu.edu.cn (Y.R.); gcy@njfu.edu.cn (C.G.); libianbian@njfu.edu.cn (B.L.); fyyw@njfu.edu.cn (Y.F.); zhaoxq402@163.com (X.Z.); 2College of Forestry, Nanjing Forestry University, Nanjing 210037, China

**Keywords:** pomegranate, SWEET gene family, gene expression, gene cloning, subcellular localization

## Abstract

Members of the sugars will eventually be exported transporter (SWEET) family regulate the transport of different sugars through the cell membrane and control the distribution of sugars inside and outside the cell. The SWEET gene family also plays important roles in plant growth and development and physiological processes. So far, there are no reports on the SWEET family in pomegranate. Meanwhile, pomegranate is rich in sugar, and three published pomegranate genome sequences provide resources for the study of the SWEET gene family. 20 PgSWEETs from pomegranate and the known *Arabidopsis* and grape SWEETs were divided into four clades (Ⅰ, Ⅱ, Ⅲ and Ⅳ) according to the phylogenetic relationships. PgSWEETs of the same clade share similar gene structures, predicting their similar biological functions. RNA-Seq data suggested that PgSWEET genes have a tissue-specific expression pattern. Foliar application of tripotassium phosphate significantly increased the total soluble sugar content of pomegranate fruits and leaves and significantly affected the expression levels of PgSWEETs. The plant growth hormone regulator assay also significantly affected the PgSWEETs expression both in buds of bisexual and functional male flowers. Among them, we selected *PgSWEET17a* as a candidate gene that plays a role in fructose transport in leaves. The 798 bp CDS sequence of *PgSWEET17a* was cloned, which encodes 265 amino acids. The subcellular localization of PgSWEET17a showed that it was localized to the cell membrane, indicating its involvement in sugar transport. Transient expression results showed that tobacco fructose content was significantly increased with the up-regulation of *PgSWEET17a*, while both sucrose and glucose contents were significantly down-regulated. The integration of the PgSWEET phylogenetic tree, gene structure and RNA-Seq data provide a genome-wide trait and expression pattern. Our findings suggest that tripotassium phosphate and plant exogenous hormone treatments could alter PgSWEET expression patterns. These provide a reference for further functional verification and sugar metabolism pathway regulation of PgSWEETs.

## 1. Introduction

Sucrose can be degraded to hexose (glucose and fructose), which is the main carbon source for plant growth and development or can act as osmoregulation substances and signal molecules. Especially in fruit crops, the accumulation of soluble sugars is not only a source of energy for fruit growth, but also determines to some extent the quality of the fruit [1]. Notably, sugar accumulation requires sugar metabolism and transport, in which sugar transporters play an integral role [2,3]. It is well known that the carbohydrate partitioning process is achieved from source to sink tissues via the element/companion cell (SE/CC) complex of the phloem [3,4]. In the leaves of many herbaceous plants, including most crops, the SE/CC complex shares very little of the plasmodesmata with adjacent cells. Thus, sucrose transport in mesophyll cells is exported to the cell wall probably by sugars will eventually be exported transporters (SWEETs), and then through sucrose transporters (SUTs/SUCs) into the SE/CC complex [4].

Currently, although the SWEET gene family has not been studied for a long time, the available findings suggest that they are involved in important physiological processes of plant growth and development (e.g., plant nectar production, seed and pollen development) by regulating the transport, distribution and storage of carbohydrate compounds [5,6]. SWEET proteins utilize intracellular and extracellular sugar concentration gradients for transmembrane movement, not proton gradients, so their sugar transport activity is not affected by environmental pH. More importantly, SWEET proteins can perform bidirectional sugar transport across the membrane driven by solute potential along concentration gradients [7,8]. Other known monosaccharide transporters (MSTs) and SUTs are coupled to H^+^ and perform unidirectional transmembrane sugar transport through intracellular and extracellular H^+^ concentration gradients [9,10]. Unlike SUTs and MSTs, which contain 12 α-helical transmembrane (TM) structural domains connected by hydrophilic loops, SWEET genes belonging to the MtN3/saliva family encode membrane proteins that typically have seven conserved TM domains linked by a PQ-loop-repeat [11]. The topology of SWEETs is different from that of SUTs and MSTs, which may be important for the different sugar transport.

In plants, phylogenetic analysis has shown that SWEETs can be divided into four clades (Ⅰ, Ⅱ, Ⅲ and Ⅳ), and genes in the same subclade have similar gene structures and functions [12,13]. Much evidence suggests that Clades Ⅰ and Ⅱ may transport glucose, Clade Ⅲ prefers sucrose, and Clade Ⅳ is an efficient fructose transporter. SWEETs act as sugar transporters involved in phloem loading during the process of sucrose metabolism (Table 1). Chen et al. [7] first demonstrated that *AtSWEET11/12* are located in phloem parenchyma, are highly expressed, and can transport sucrose from leaves to vascular bundles. *AtSWEET17* is highly expressed in the vacuolar membrane in *Arabidopsis* leaves and root and is responsible for fructose bidirectional transport [14]. Members of the SWEET family are frequently involved in plant reproduction and development (Table 1). *AtSWEET9* is specifically expressed in parenchyma cells of nectary, while mutant *atsweet9* results in reduced nectar production [15]. *Ossweet14* mutant knockouts caused less plump seeds and delayed growth, suggesting that *OsSWEET14* plays a role in grain filling [16]. Meanwhile, *OsSWEET1a*, *OsSWEET2a*, *OsSWEET3a*, *OsSWEET4*, *OsSWEET5* and *OsSWEET15* were highly expressed in plant flowers and panicles, indicating that these transporters also play a certain role in rice reproductive development [17]. In addition, SWEETs are also involved in ion transport, leaf senescence, plant-pathogen interaction and abiotic stress [18,19,20,21].

Pomegranates are grown in different subtropical and tropical microclimatic zones around the world, such as Iran, California, Turkey, Egypt, Italy, India, Chile, Spain, and China. The market demand for pomegranate products such as pomegranate and its by-products juice, jam, and pomegranate wine are steadily increasing due to the growing consumer interest in the potential benefits of pomegranate and its phytochemicals. Pomegranate has been used since ancient times for the prevention and treatment of several diseases, has strong antioxidant activity, and is rich in anthocyanins, tannins and unique punicalagin. Studies have shown that pomegranate fruit and its juice, extracts and oils are able to exert anti-inflammatory, anti-proliferative and anti-tumor effects by modulating multiple signaling pathways [31,32,33,34].

At present, studies on pomegranate sugar metabolism have so far focused on the determination of soluble sugar components and contents, and there is a lack of research on the regulatory mechanism of sugar metabolism. Pomegranate juice is classified into three categories based on its soluble solids (TSS) and titratable acid (TA) content; juice with TSS above 13% and TA below 0.7% is classified as sweet pomegranate, juice with TSS range of 12–13% and TA of 0.7–1.8% is classified as sweet and sour pomegranate, and juice with TSS below 12% and TA above 1.8% is classified as sour pomegranate [35]. In addition, the flower buds of pomegranate are distinguished as fertile and abortive flowers. Flower development consists of two stages including (i) the transition from vegetative to reproductive development and (ii) organogenesis, both of which are regulated by a number of well-characterized key genes leading to abortive flowers [36]. Interestingly, it has been found that sugar/hormone interactions, or direct effects on the transcription of genes that regulate flower induction, ensure flower bud formation [37,38,39].

Studying the function and regulatory mechanism of the pomegranate SWEET gene family can help regulate the pattern of sugar metabolism as well as fruit, leaf and flower development. The completion of pomegranate whole genome sequencing and the publication of data provide important data to support the study of pomegranate gene function [40,41,42]. Using bioinformatics methods, we identified 20 PgSWEETs, and analyzed their physical and chemical properties, phylogenetic relationships, gene structures and RNA-seq data. We also explored the expression patterns of PgSWEETs after foliar application of tripotassium phosphate and plant growth hormone regulators. Finally, gene cloning, subcellular localization and transient expression of tobacco of *PgSWEET17a* were analyzed to provide reference for studying its biological functions in pomegranate sugar transport.

## 2. Results

### 2.1. PgSWEET Gene Family Members Identification and Sequence Analysis

We identified 20 pomegranate SWEETs according to the Pfam database (ID: PF03083). Physicochemical properties of the genes were analyzed through online software Protparam (Table 2). Analysis showed that length of amino acid of 20 PgSWEETs were ranged from 213 (*PgSWEET7a*) to 448 (*PgSWEET1e*) aa. Molecular weight was from 23,370.82 (*PgSWEET17b*) to 50,147.21 (*PgSWEET1e*) ku. The pI of 2 PgSWEETs (*PgSWEET17b* and *PgSWEET12*) were less than 7, which were more acidic, while others were more than 7, which were more basic.

### 2.2. Phylogenetic Tree Analysis

We identified 20 PgSWEETs from pomegranate, then reconstructed one gene tree with known AtSWEETs and VvSWEETs for further study SWEET family evolutionary relationship (Figure 1). Apparently, SWEETs from three species were split into four clades (Ⅰ, Ⅱ, Ⅲ and Ⅳ, Figure 1). Among them, Clade Ⅰ has 7 PgSWEETs, Clade Ⅱ has 4 PgSWEETs, Clade Ⅲ has 5 PgSWEETs, and Clade Ⅳ has 4 PgSWEETs. There is a germline-specific clade (*PgSWEET1a*-*1e*) of pomegranate in Clade Ⅰ.

### 2.3. PgSWEETs Gene Structure and Phylogenetic Tree Analysis

To investigate the phylogenetic relationships of the PgSWEET gene family, we reconstructed the evolutionary tree with 20 pomegranate gene sequences by IQ-tree (Figure 2A). Apparently, phylogenetic analysis showed robust support for Clades Ⅰ, Ⅱ and Ⅳ, and less support for Clade Ⅲ. Five homologous pairs were formed among these 20 sequences, including *PgSWEET1a* and *PgSWEET1b*, *PgSWEET1c* and *PgSWEET1d*, *PgSWEET7c* and *PgSWEET7d*, *PgSWEET16a* and *PgSWEET16b*, and *PgSWEET17a* and *PgSWEET17b*. All pairs, except *PgSWEET17a* and *PgSWEET17b*, possessed an approval rate of 100, indicating a very strong phylogeny and a close association with each other.

We utilized the online software GSDS for investigating the PgSWEET gene structural features (Figure 2B). Structural analysis showed that most genes have 5 or 6 exons, while *PgSWEET7a* only has 3 exons and *PgSWEET1e* has 7 exons. The most obvious difference among these genes structure is the length of introns. Genes in Clade Ⅰ have the longest introns, and those in Clade Ⅲ have shorter introns. The length of introns grows gradually as the phylogenetic tree branches from the outer to the inner branches. Combined with phylogenetic tree and Scaffold location analysis, *PgSWEET1b* and *PgSWEET1c*, *PgSWEET7a* and *PgSWEET7b*, and *PgSWEET16a* and *PgSWEET16b* were found to have tandem duplication.

### 2.4. PgSWEETs Gene Expression Analysis

To further investigate the gene expression divergence among different tissues, we downloaded RNA-seq data from NCBI for pomegranate (Table 3, Figure 3). Among the 20 PgSWEET gene expressions in different tissues, five genes (*PgSWEET1a*, *1b*, *3*, *8* and *17b*) were not expressed or minimally expressed in various tissues. *PgSWEET1d* was highly expressed only in the root of ‘Dabenzi’. Four genes (*PgSWEET5*, *7a*, *9* and *11*) showed similar expression patterns, and in the flower of ‘Dabenzi’, mixed samples of ‘Black127′ and ‘Nana’ showed higher expression levels. In particularly, *PgSWEET7a* and *11* showed the highest transcript levels in the flower of ‘Dabenzi’. *PgSWEET1c* and *1d* displayed the highest expression levels in the root. Only three genes *PgSWEET2*, *15* and *16b* displayed expression in all tissues. Among them, *PgSWEET2* expression in all tissues were all higher, indicating that *PgSWEET2* may play a multifaceted function in the growth and development of pomegranate.

### 2.5. Changes in Soluble Sugar Concentration and PgSWEETs Expression Pattern of Pomegranate Fruit and Leaf after Tripotassium Phosphate Treatment

When the pomegranate fruit enters the color turning period (4–5 weeks before harvest), the growth of the fruit and the accumulation of sugar require nutrients, and the demand for fertilizer increases at this stage, which is the most demanding stage in the whole growth process. At this time, in addition to basic and reasonable fertilization, proper foliar application can supplement sufficient nutrients to improve fruit growth, increase sweetness, and enable successful coloring. Therefore, we started foliar spraying of “Big Seed” pomegranate with tripotassium phosphate solution from September 20 (pomegranate fruit color change phase) to 10 October. Compared with the control group, the results showed that fruit TSS/TA, fruit and leaf total soluble sugar contents were significantly enhanced after foliar spraying with tripotassium phosphate (Figure 4A–C).

To investigate the effect of foliar application of tripotassium phosphate on the fruit sugar transporter, the transcript levels of 16 PgSWEET genes were determined (Figure 4D). The expression of SWEETs in pomegranate fruits was generally low from the color turning stage to the ripening stage, and the expression of the same gene varied greatly at different times. Overall, the expression of SWEETs in fruits was relatively high on 20 September, and the expression of nine genes (*PgSWEET1c*, *3*, *5*, *7a*, *7b*, *9*, *11*, *15* and *17a*) was the highest compared with other periods. Foliar application of tripotassium phosphate also altered the expression pattern of SWEETs in fruits, with the highest expression of *PgSWEET2* and *16b* in the treatment group on 30 September and the highest expression of *PgSWEET1e* and *16a* in the treatment group on 10 October. In addition, foliar fertilization decreased the expression of *PgSWEET10* compared to the control group.

To explore the functional differences of sugar transport in source organs after foliar application of tripotassium phosphate compared with the control, we determined the expression of 16 PgSWEET family genes in the treated and control groups of leaves and drew a clustering heat map (Figure 4E). There were differences in the expression of PgSWEETs between different periods and treatments. It was clearly seen that the expression of PgSWEETs was significantly higher in the treatment group on 10 October, where the expression of 13 genes (*PgSWEET1c*, *1e*, *2*, *5*, *7a*, *7b*, *9*, *10*, *12*, *16a*, *16b*, *17a* and *17b*) was higher than in other stages and the expression of *PgSWEET11* was also at a higher level. On 20 October, gene expressions were lower in both control and treated groups except for *PgSWEET2*, *3* and *5*, and it is speculated that leaves started to enter the senescence stage. There were also significant differences in the expression of PgSWEETs between the treatment and control groups on 30 September. For example, after foliar application of tripotassium phosphate, the expression levels of *PgSWEET1c* and *17b* were significantly higher than those of the control group, while the expression levels of *PgSWEET1e*, *2*, *9*, *11*, *16a* and *17a* were significantly lower than those of the control group.

### 2.6. Changes in Buds Total Soluble Sugar Contents and PgSWEETs Expression Pattern of Bisexual Flower and Functional Male Flower after Hormone Treatment

We measured the buds sugar content at different periods under hormone treatment and found that their sugar content was affected by hormones (Figure 5A,B). Then, the expression of some PgSWEETs was measured to explore their changes during this process (Figure 5).

The qPCR results showed that the expression of *PgSWEET1c* and *PgSWEET2* in CK was higher in bisexual flowers than in functional male flowers. The expression of *PgSWEET1e* and *PgSWEET15* was lower in bisexual flowers than in functional male flowers. The expression of *PgSWEET16b* was significantly higher in the P3 stage of bisexual flowers than in functional male flowers, and the expression of *PgSWEET17a* was lower in the P1 stage of bisexual flowers than in functional male flowers. The expression of *PgSWEET17a* was lower in the P1 stage of bisexual flowers than in functional male flowers. In the CK group of bisexual flowers, *PgSWEET1c*, *1e*, *15* and *17a* expression was highest in the P2 stage, and *PgSWEET2* and *16b* were highest in the P3 stage. In functional male flowers of the CK group, *PgSWEET1c*, *1e*, and *16b* expression was highest in the P2 stage, *PgSWEET2* and *17a* expression continued to decrease, and *PgSWEET15* continued to increase.

Under IBA treatment, *PgSWEET1c* and *16b* were down-regulated in both types of flowers compared to the control. *PgSWEET1e*, *2* and *15* were up-regulated in bisexual flowers due to exogenous IBA treatment and down-regulated in functional male flowers. *PgSWEET17a* was up-regulated in all three stages in bisexual flowers and up-regulated in functional male flowers at the P1 and P3 stages, whereas it was down-regulated at the P2 stage.

Under 6-BA treatment, *PgSWEET1c*, *2* and *16b* were down-regulated in expression at all three stages in bisexual flowers compared with control, whereas *PgSWEET17a* was up-regulated. *PgSWEET1e* and *15* expression were down-regulated at the P1 stage and up-regulated at the P2 and P3 stages in bisexual flowers. Although the expression of the six PgSWEETs in functional male flowers under 6-BA treatment was variable, it could be found that the expression of all six PgSWEETs was significantly lower than control at the P2 stage, a critical period for sterile flower abortion.

Under PP_333_ treatment, the expression of all six PgSWEETs was down-regulated in functional male flowers compared to control, except for *PgSWEET17a*, which was down-regulated at the P1 and P3 stages. PgSWEETs expression changes were more complex in bisexual flowers compared with functional male flowers. In bisexual flowers, *PgSWEET1c*, *2*, and *16b* were all down-regulated, and *PgSWEET15* was up-regulated. *PgSWEET1e* and *17a* were up-regulated at the P1 and P2 stages and down-regulated at the P3 stage compare with control.

Overall, under exogenous hormone treatment, *PgSWEET1c*, *2* and *16b* expression may be down-regulated, and *PgSWEET1e*, *15* and *17a* expression was up-regulated in bisexual flowers compared with control. Six PgSWEETs were down-regulated in functional male flowers at all three stages or critical periods.

### 2.7. Cloning and Structural Analysis of PgSWEET17a

The AtSWEET17 protein sequence of *Arabidopsis thaliana* was utilized as the probe sequence, and the homologous gene sequence obtained was named *PgSWEET17a* by blastp comparison (http://cucurbitgenomics.org/blast, accessed on 26 August 2021) in the pomegranate genome database. Amplification primers were designed according to the known CDS sequence, and PCR amplification was performed using cDNA mixes from pomegranate flower, leaf, fruit and skin tissues as templates, and the amplified products were analyzed by agarose gel electrophoresis, recovered, cloned and sequenced. The cDNA of the *PgSWEET17a* gene was obtained by cloning with a full length of 798 bp and a CDS sequence of 798 bp (Figure 6A) Pfam database search revealed that this protein has the MtN3/saliva conserved structural domain (Figure 6B).

Using physicochemical property analysis of the sequence, the results showed that PgSWEET17a protein encodes 265 amino acids and the molecular formula of the encoded product is C1383H2130N330O369S10 (Figure 6B), which is predicted by online software Plant-mPLoc to be localized on the cell membrane. It has a molecular weight of 29.60 kD, an aliphatic index of 112.87, a grand average of hydrophilicity of 0.532, a theoretical isoelectric point (pI) of 6.57, and an instability index of 44.44.

### 2.8. PgSWEET17a Conservative Domain Analysis

Plant SWEET proteins belong to the MtN3/saliva family and generally contain seven TMs, of which TM4 is poorly conserved and acts as a linker, forming a 3-1-3 structure that forms two MtN3/saliva structural domains, each containing three TMs. In contrast, prokaryotes have only one MtN3/saliva structural domain, and it is speculated that eukaryotic SWEETs evolved through the internal replication (replication or horizontal gene transfer fusion) of the 3 TMs [8,45]. From Figure 7, PgSWEET17a apparently has 7 TMs.

To further study the conserved domain of PgSWEET17a MtN3/saliva, PgSWEET17a and PgSWEET17b from pomegranate and 13 SWEET proteins from grape, rice, tomato and *Arabidopsis* were aligned to detect conserved domains (Figure 7). PgSWEET17b is a homologous sequence of PgSWEET17a, while it has multiple base substitutions or deletions in conserved regions, which could result in neo- or sub-functionalization. The alignment exhibited that the residues G9, G12, N13, P23, T41, F42, P58, Y59, Y72, G73 and Y98 on THB1 and G149, P163 and L183 on THB2 were completely conserved. Besides, more than 90% of SWEETs have I45, T86, G89 and G91 on THB1 and S175, V176, P180, S184, Y225 and Y228 residues. Furthermore, the prolines (P) in TM1, TM2, TM5 and TM6 were conserved in 15 SWEETs, and only PgSWEET17b had a glutamine (Q)-substitution in the TM6, which may be responsible for the loss of activity in PgSWEET17b [46]. Similarly, two conserved residues N88 and N211 in TM3 and TM7, respectively, were vital for AtSWEET1 activity [46], and were reserved in 14 SWEETs except for PgSWEET17b.

### 2.9. PgSWEET17a Subcellular Localization Analysis

Predictive analysis of the subcellular localization of PgSWEET17a protein using the online tool Plant-mPLoc showed that it was localized to the cell membrane. To verify the above predicted results, the homologous recombinant *PgSWEET17a*-GFP fusion protein was transiently expressed in tobacco leaves, and the results showed that PgSWEET17a was localized to the cell membrane (Figure 8). It is presumed to be involved in sugar transport.

### 2.10. PgSWEET17a Cis-Acting Elements Analysis

The results are shown in Table 4, where the *PgSWEET17a* sequence contains cis-acting elements associated with abiotic stresses and hormones. SREATMSD, ABRE, ARE, AuxRR-core, CGTCA-motif and CGTCA-motif are cis-acting elements associated with the sugar signal, abscisic acid responsiveness, anaerobic induction, auxin responsiveness, MeJA-responsiveness, light responsiveness, low-temperature responsiveness and drought-inducibility, respectively.

### 2.11. PgSWEET17a Tobacco Transient Expression Analysis

We compared the expression of *PgSWEET17a* in pomegranate fruits, leaves, buds and flowers (Figure 9B). *PgSWEET17a* expression was found to be significantly higher in leaves, buds and flowers (*p* < 0.05) than in fruits. To investigate the effect of *PgSWEET17a* on sugar concentration, the constructed pBI121-*PgSWEET17a* vector was infiltrated into tobacco (NB), and tobacco leaves were harvested on the third (3d), fourth (4d) and fifth (5d) days after dark incubation for *PgSWEET17a* expression and soluble sugar content determination (Figure 9A). Figure 9D shows that the expression of *PgSEET17a* was significantly higher on days four (*p* < 0.01) and five (*p* < 0.05) than in wild-type (WT) leaves, and there was no difference between leaves of WT and pBI121, and WT and 3d, indicating that *PgSWEET17a* was significantly induced in tobacco after infestation. Interestingly, fructose contents were significantly increased in leaves on the 3d, 4d and 5d and sucrose and glucose contents were both remarkedly decreased (Figure 9C).

In addition, we identified the expression of the sugar metabolism related enzymes *SS*, *SPS* and *INV* genes in tobacco leaves. The *SS* transcript level, was significantly lower than that of WT on all three days (*p* < 0.01, Figure 9E). *SPS* gene expression in leaves on the 3d (*p* < 0.05) and 5d (*p* < 0.01) were significantly lower than WT (Figure 9F). *INV* expression levels on the 3d (*p* < 0.05) and 4d (*p* < 0.01) were both significantly higher than WT, while on the 5d it was significantly lower than WT (Figure 9G). There was no difference in expression of these genes between WT and pBI121(Figure 9). Overall, with the overexpression of *PgSWEET17a*, the expression of *SS* and *SPS* was induced to be down-regulated in tobacco leaves, while the *INV* expression was significantly induced to be up-regulated at first and significantly decreased at a later stage.

## 3. Discussion

In higher plants, soluble sugars are mainly in the form of sucrose, glucose and fructose [47]. Sugar transport and distribution play an important role in promoting plant growth and development and crop yield improvement. The efficient regulation and distribution of sugar transport from source to sink organs through the phloem in plant tissues is one of the major determinants of plant growth [48]. Members of the SWEET family regulate the transport of different sugars through the cell membrane and control the sugars distribution inside and outside the cell. Currently, SWEETs genes are widely found in higher eukaryotes, as well as in protozoa, metazoans, fungi, bacteria and archaea [22]. The diversity of SWEETs in higher plants also reflects their different functions in growth, development and physiological processes, including pollen, seed and fruit development and nectar production [9,13,23]. The SWEETs family of genes has been found in many plants, such as 17, 21, 29, 17, 23, 13, 52, and 16 in *Arabidopsis*, rice, tomato, grape, sorghum, tea, soybean, and litchi, respectively [8,12,13,49,50,51,52,53]. Previous reports on the pomegranate SWEET family were missing. Now, three pomegranate genomes including ‘Dabenzi’, ‘Tunisa’ and ‘Taishanhong’ have been reported [40,41,42], which provides a reference for the identification and analysis of pomegranate SWEET genes and the functional exploration in this study.

Twenty candidate PgSWEET family members were identified and analyzed. The analysis showed that there were tandem duplications of PgSWEET genes. It has been suggested that the expansion of SWEET genes in dicots is mainly the result of recent duplication events, especially tandem duplication [54]. Based on the phylogenetic tree reconstructed using pomegranate, *Arabidopsis* and grape SWEETs, the analysis showed that the PgSWEETs family was divided into four subfamilies, namely Clade I, II, III and IV. Different PgSWEET genes have different structures, with little variation in the number of exons and large variation in the length of introns, and a tendency to grow as the evolutionary tree branches from the outside to the inside. The last common ancestor of angiosperm SWEETs was speculated to have 6 exons and gene size variation was presumed to be caused by intron insertion [54]. Most SWEET genes in pomegranate, *Arabidopsis* and grape have a 1:1:1 relationship in number, which may be due to a γ WGD (whole genome duplication) event in angiosperms [40].

A specific expansion subclade consisting of five PgSWEETs was present in Clade I, and traces of *PgSWEET1a* and *1b* expression were found based on transcriptomic results, which may be due to pseudogenization as a result of duplication events [55]. *PgSWEET1c* was highly expressed in roots, as well as expressed in both bisexual and functional male flower buds. *PgSWEET1d* was only higher expressed in roots. It was speculated that *PgSWEET1c* and *1d* were important in glucose transport from roots [8]. *PgSWEET1e* was induced with the development of functional male flowers, implying that it provides nutrition to the developing gametophyte or nectaries [8]. *AtSWEET15* in *Arabidopsis* has been validated to be associated with plant senescence and to respond to adversity stress to maintain normal plant growth [21,56]. The induction of some genes involved in plant defense mechanisms is an integral part of the leaf senescence program [57]. *PgSWEET16s* and *17s* have high sequence similarity and belong to the Clade IV, a branch mainly involved in fructose transport. *PgSWEET16b* was highly expressed in leaves and *PgSWEET17a* was mainly expressed specifically in leaves, skins and roots, suggesting that they regulate fructose content in these tissues [14,20,30].

Indeed, foliar application of tripotassium phosphate at the time of pomegranate fruit color turning stage increased the fruit soluble sugar content. To preliminarily investigate the mechanism of regulation of sugar metabolism by tripotassium phosphate in pomegranate, we determined the expression of the PgSWEETs family in pomegranate leaves and fruits. On September 20, all genes were highly expressed except for *PgSWEET1e*, *2*, *10* and *16a*. Later, in the control group, most PgSWEETs had low transcript levels, while *PgSWEET1e*, *2*, *10* and *16a* were expressed at high levels, and *PgSWEET9*, *16b* and *17b* were expressed at slightly higher levels on 10 October, suggesting that these genes may play an important role in determining fruit quality and yield at fruit ripening [53]. Foliar spraying of tripotassium phosphate significantly increased *PgSWEET2*, *5*, *9*, *12*, *16b* and *17b* compared to control on 30 September, and increased *PgSWEET1e* and *16* expression on 10 October in fruit. These genes produced high expressed levels in response to tripotassium phosphate in fruit. In pomegranate leaf, *PgSWEET1e*, *2*, *9*, *11*, *16a* and *17a* expression in treatment group was significantly lower than control on September 30, and most genes expression was significantly higher than control except for *PgSWEET3* and *15* on 10 October. These implied the spatiotemporal expression specificity of PgSWEETs. PgSWEETs had high expression levels in response to tripotassium phosphate in leaf. *PgSWEET15* in leaf after tripotassium phosphate treatment was significantly decreased, which might be due to the delayed leaf senescence [21]. These suggest that SWEETs are involved in regulating the redistribution of soluble sugars within pomegranate tissues in response to tripotassium phosphate to influence plant growth.

Sugar regulates juvenile and floral signaling through energy sources, osmoregulation, and signaling molecules, and sugar signaling has been proven to be important for coordinating developmental transitions in interaction with phytohormones [38]. ABA is a floral induction inhibitor whose components of the synthesis and transduction pathway are allelic to mutations that affect sugar signaling [58,59]. Meanwhile, glucose can enhance ABA concentration and ABA-biosynthesis-related gene expression [60]. Ethylene has regulatory effects on juvenile development and flower induction [61]. In addition, ethylene can affect the sensitivity of plants to sugars, and the application of ethylene precursors can reduce the sensitivity of plants to glucose [62]. It was clear that the soluble sugar content of flower buds was significantly induced after hormone treatment. We determined the expression of several important PgSWEETs and found that hormone treatment has an effect on the sugar transporter protein. Six PgSWEETs were differentially expressed in bisexual and functional male flowers and showed different trends in response to different hormones, suggesting that these genes may be involved in stress response or in the response process of different phytohormone signal transduction in pomegranate development.

It was shown that most SWEET proteins are localized on the plasma membrane and may be the main sites for regulating sugar flux [63,64]. SWEET proteins have been identified as bidirectional transporter proteins that regulate cellular uptake and efflux of sugars, but there are differences in substrate specificity among SWEETs [7,65]. In *Arabidopsis*, *AtSWEET17* is located in the vacuole membrane and can regulate changes in fructose levels in leaves. Under low nitrogen conditions, large amounts of sugars accumulate in plant tissues and large amounts of monosaccharides (glucose and fructose) accumulate in the vesicles via the vacuole monosaccharide transporter TMT, from where *AtSWEET17* apparently exports a portion of fructose to the cytoplasm; this function is impaired in the mutant *sweet17*, leading to a specific accumulation of fructose in the tissues [14]. In higher plants, the rapid accumulation of soluble sugars can cause metabolic changes. In wild-type tobacco, sucrose is mainly distributed in the cytoplasm, with up to 98% hexose in the vacuole, and in transgenic tobacco, the hexose content is higher than in wild-type plants, again with 97–98% hexose stored in the vacuole; in short, the vacuole in the leaves actively take up the transporters of glucose and fructose at high concentration gradients [66]. In this study, overexpression of *PgSWEET17a* caused an increase in fructose content in transgenic tobacco, while both sucrose and glucose content decreased. It is speculated that *PgSWEET17a* is involved in the transport of fructose, sucrose and glucose. In addition, *PgSWEET17a* expression level was affected by exogenous hormones in pomegranate flower buds. We found that the expression of *PgSWEET17a* was significantly higher in bisexual flower buds than in functional male flower buds during the critical period of sterile flower abortion, i.e., the P2 stage. It is hypothesized that *PgSWEET17a* plays a role in the formation of fertile flowers in buds.

Sugar metabolism is influenced by a variety of enzymes and their related genes. During fruit development, sucrose transport from source tissues to sink organs through the phloem requires SS, INV and SPS to complete the conversion between sucrose, fructose and glucose, and to create a sucrose concentration gradient between source and sink to drive sugar transport. SS plays an important role in plant cellulose synthesis, starch synthesis and sucrose transport, and its role in overexpressing *OsSUS3* and *AtSUS* in plants and citrus fruits has been verified [67,68,69]. Furthermore, sucrose inversion catalyzed by INV requires two molecules of ATP, while only one molecule of PPi is consumed via SS, which can be recycled, making the SS pathway more energy-efficient [70]. Related evidence suggests that overexpression of *INV* in potato plants consumes too much ATP, and although the up-regulated *INV* promotes sucrose inversion, it also limits starch accumulation in the stem mass [71].

## 4. Material and Methods

### 4.1. Identification and Sequence Analysis of PgSWEETs

HMM model files were constructed based on the SWEET gene family sequence alignment file (PF03083) downloaded from the Pfam database, and protein sequences containing the conserved structural domain of SWEET (E-value ≤ e^−10^) in the pomegranate genome file [40] were screened using SelectHMM (https://github.com/Redpome/SelectHMM, accessed on 1 February 2022). Then, prediction of physicochemical properties of PgSWEET protein sequences using the online software Protparam (https://web.expasy.org/protparam/, accessed on 1 February 2022).

### 4.2. Phylogenetic Tree Construction

The protein sequences of the screened PgSWEETs were aligned with *Arabidopsis* and grape SWEETs using MAFFT7.487 [72], and the protein sequence alignment was transformed into the corresponding codon alignment using pal2nal script [73] to construct a phylogenetic tree using iqtree [74]. The evolutionary tree is visualized using figtree software (http://tree.bio.ed.ac.uk/software/figtree/, accessed on 1 February 2022).

### 4.3. PgSWEETs Gene Structure Analysis

Gene structure information of the PgSWEET gene family, including introns, exons and upstream and downstream sequences, was obtained from the pomegranate genome annotation file, and the results were presented using the online software GSDS (http://gsds.cbi.pku.edu.cn, accessed on 1 February 2022).

### 4.4. PgSWEET Gene Family RNA-Seq Analysis

To analyze expression patterns of PgSWEETs in different pomegranate tissues and organs, the published transcriptome data (Table 3) were downloaded from NCBI (http://www.ncbi.nlm.nih.gov/, accessed on 1 February 2022). The transcriptomic data were calculated and analyzed using Kallisto v0.44.0 software (CA, USA), and the resulting values were transformed into Log_2_(TPM+1) and finally clustered for heat mapping using pheatmap in R [75].

### 4.5. Plant Material and Tripotassium Phosphate Foliar Fertilization

The pomegranate fruits and leaves were collected from September to October 2020 in Hu Gang Pomegranate Garden, Liuhe District, Nanjing, Jiangsu Province. The treatments for the field trial were as follows: foliar sprays were applied to uniformly growing pomegranate trees, with prue water as the control group (CK) and 500 ppm concentration of tripotassium phosphate (K_3_PO_4_) as the treatment group. Five pomegranate trees were selected for each treatment. Spraying began on 20 September 2020 (pomegranate fruit color turning stage), and was applied at 10-day intervals for a total of three sprays. The level of spraying was until the leaves dripped on both sides. The pomegranates were picked before each spraying, 4 times in total. Pomegranate fruit and leaves of consistent growth and free of pests and diseases were picked from each tree in all directions. The last picking was done on 20 October (fruit ripening period). The fruit seeds were mixed well after each picking, treated with liquid nitrogen and stored in −80 °C refrigerator for backup. (No permission was required for sample collection.)

### 4.6. Plant Material and Exogenous Hormone Treatments

“Taishanhong” pomegranate was selected as the experimental cultivar. Three plant growth regulators (CK: pure water, 100 mg/L 6-BA, 20 mg/L IBA and 1000 mg/L PP_333_) were selected and sprayed on the leaves as follows: three sprays were applied on 5 October 2019 (dormant period), 15 April 2020 (spring leaf expansion period) and 10 May (initial flowering to full flowering period). Flower buds of bisexual and functional male flowers of pomegranate at different stages of development were collected. Flower buds were divided into three stages according to their longitudinal diameter: 3–5 mm (P1), 5.1–13 mm (P2) and 13.1–25 mm (P3). (No permission was required for sample collection.)

### 4.7. Total Soluble Sugar Content Determination

Total soluble sugar of fruits, leaves and buds was determined using the anthrone-sulfuric acid colorimetry method [76].

### 4.8. RNA Isolation and Gene Clone

The reagents for the test included laboratory-preserved overexpression vector pBI121, DHα (Beijing Prime Tech Biotechnology Co., Ltd., Beijing, China), GV3101 (Shanghai Viearth Biotechnology Co., Ltd., Shanghai, China), One-Step Cloning Kit and 2×Taq Plus Master Mix enzyme (Nanjing Novizan Biotechnology Co., Ltd., Nanjing, China), Polysaccharide Polyphenol Plant RNA Extraction Kit (Beijing Tiangen Biochemical Technology Co., Ltd., Beijing, China), Reverse Transcription Kit (Takara Company, Tokyo, Japan), and PrimeScriptTM RT reagent Kit with gDNA Eraser (Takara Company), DNA Maker and Glue cutting Recovery Kit (Beijing Qingke Biotechnology Co., Ltd., Beijing, China). The primers used in the experiments were synthesized by Shanghai Bioengineering Compony (Table 5).

The PCR amplification system was: 25 μL 2×Taq Plus Master Mix; 1 μL forwardprimer; 1 μL reverseprimer; 2 μL DNA template; 21 μL nuclease-free ddH2O. PCR program settings were: 95 °C: 3 min; 95 °C: 15 s; 52 °C: 45 s; 72 °C: 1 min; for a total of 35 cycles; 72 °C: 5 min; 4 °C: save. The PCR products were separated by agar gel electrophoresis, the target fragments were recovered by gel cutting, and then the CDS sequences of *PgSWEET17a* gene were obtained by amplification, ligation and transformation steps, and translated into amino acid sequences using Translate online software (https://web.expasy.org/translate/, accessed on 9 December 2021).

### 4.9. PgSWEET17a Bioinformatics Analysis

The basic physicochemical properties of the PgSWEET17a protein sequence were analyzed using the online tool ProtParam (https://web.expasy.org/protparam/, accessed on 9 December 2021). Its subcellular localization was predicted using the online tool Plant-mPLoc (http://www.csbio.sjtu.edu.cn/bioinf/plant-multi/, 9 December 2021).

Homologous protein search was performed using Blastp (http://www.ncbi.nlm.nih.gov/BLAST/, accessed on 9 December 2021). Multiple sequence alignment and visualization of amino acid sequences were performed using DNAMAN and WebLogo, respectively.

### 4.10. PgSWEET7a Subcellular Location Analysis

The correct sequence *PgSWEET17a* obtained by cloning was ligated to the pBI121 vector and the carbon end was fused to green fluorescent protein (GFP). The recombinant plasmid was then introduced into Agrobacterium tumefaciens GV3101 by the freeze-thaw method. The pBI121-GFP was used as a control, and both the recombinant plasmid and the control plasmid were infiltrated with young leaves of *N. benthamiana* by the Agrobacterium-mediated method. After dark and light cultivation for 24 h respectively, fluorescence images were collected under a confocal microscope.

### 4.11. PgSWEET17a Cis-Acting Elements Analysis

A 1500 bp promoter sequence upstream of the initial codon was extracted from the PgSWEET17a genomic sequence and used to predict cis-regulatory elements using PlantCARE (http://bioinformatics.psb.ugent.be/webtools/plantcare/html/, accessed on 9 December 2021).

### 4.12. Real-Time Fluorescence Quantitative Analysis (qRT-PCR)

qRT-PCR was used to determine the expression pattern of PgSWEETs in pomegranate fruits, leaves, flower buds and tobacco leaves. The fluorescent dye used was the BioEasy Master Mix Plus (SYBR Green, Beijing Bioteke Biotechnology Co., Ltd., Beijing, China), and the reaction procedure was: 95 °C: 3 min; 95 °C: 30 s; 60 °C: 15 s; 72 °C: 20 s, 40 cycles from step 2 to step 4. The pomegranate PgActin and NtActin genes were used as an internal reference and three biological replicates were performed for each treatment. The data were quantified using the 2^−ΔΔCt^ method. Primers used for qRT-PCR are shown in Table 2 and Appendix A.

### 4.13. Agrobacterium Infiltration

The *PgSWEET17a* recombinant plasmid with GFP tag was transferred into Agrobacterium tumefaciens GV3101, activated and inoculated in 50 mL LB liquid medium, incubated in a shaker at 28 °C, 220 r/min for 12 h. After removal, the bacterium was collected by centrifugation at 4 °C, 4000 rpm/min for 10 min, and a resuspension solution was prepared (10 mmol/L MES + 10 mmol/L MgCl_2_·6H_2_O + 100 mmol/L AS, pH 5.6) 10 The bacteria were washed and resuspended at room temperature for 2 h. The bacteria were injected into tobacco leaves, sampled after infestation, and stored at −80 °C. (Tobacco seeds were kept in our laboratory).

## 5. Conclusions

In this study, we identified 20 pomegranate SWEET genes. Phylogenetic tree analysis revealed that PgSWEETs were divided into four subgroups and that genome-wide replication events and tandem replication together contributed to the evolution of the SWEET gene family. Based on the available transcriptomic data, we identified certain SWEETs that play important roles in pomegranate growth and development. Foliar application of tripotassium phosphate increased the sugar content of pomegranate leaves and fruits, and hormone treatment also affected sugar metabolism in pomegranate bisexual and functional male flowers. qRT-PCR analysis showed that PgSWEETs responded to both tripotassium phosphate and hormone treatments to varying degrees. Subcellular localization revealed that PgSWEET17a protein was localized on the cell membrane. The transient expression technique revealed that *PgSWEET17a* is involved in the transport of fructose, sucrose and glucose, with positive regulation of fructose and negative regulation of sucrose and glucose.

## Figures and Tables

**Figure 1 ijms-23-02471-f001:**
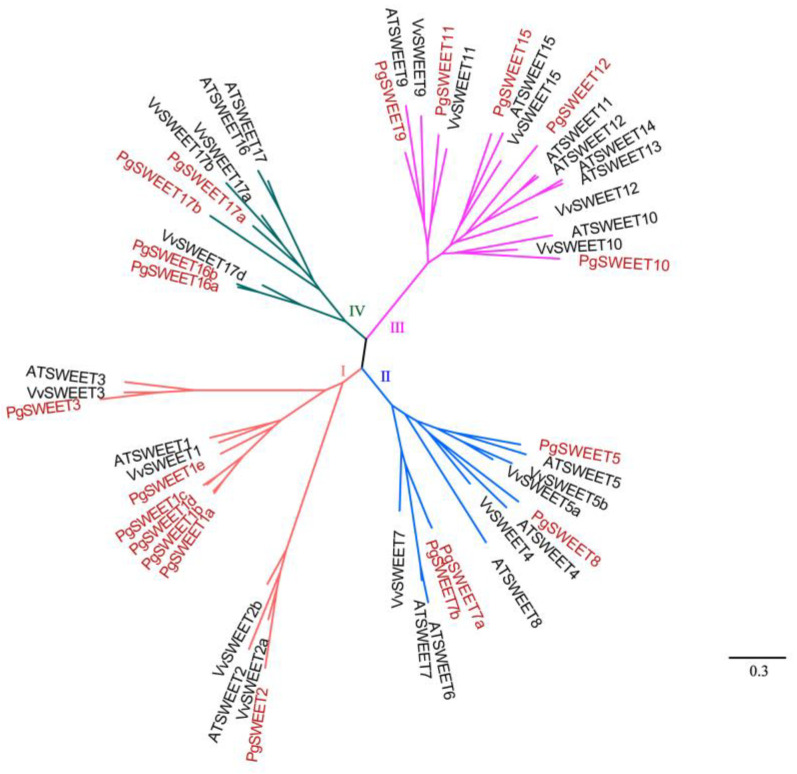
Phylogenetic tree of the SWEET gene family in pomegranate, *Arabidopsis* and grape.

**Figure 2 ijms-23-02471-f002:**
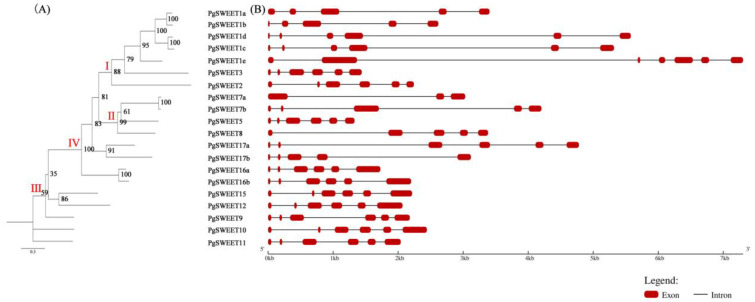
Phylogenetic tree (**A**) and gene structures (**B**) of the pomegranate SWEET gene family.

**Figure 3 ijms-23-02471-f003:**
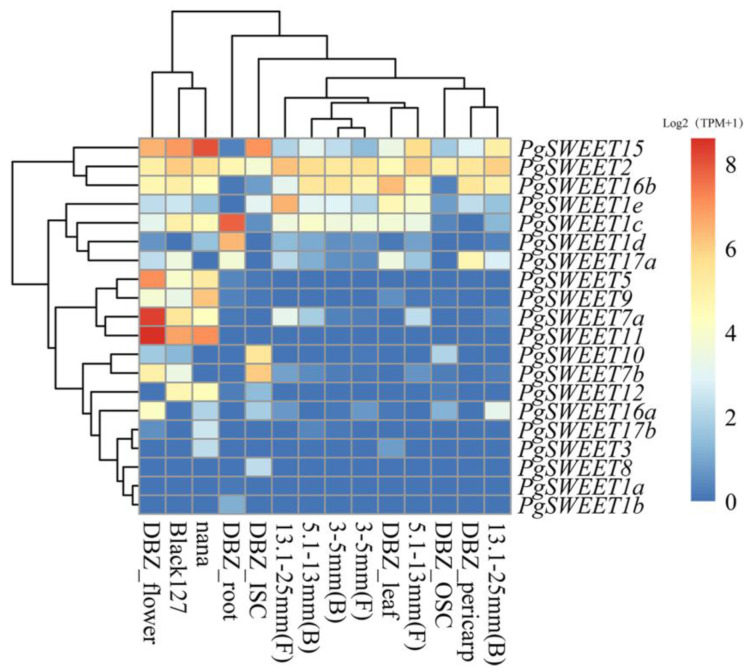
Expression heatmap of SWEET genes in different tissues of pomegranate.

**Figure 4 ijms-23-02471-f004:**
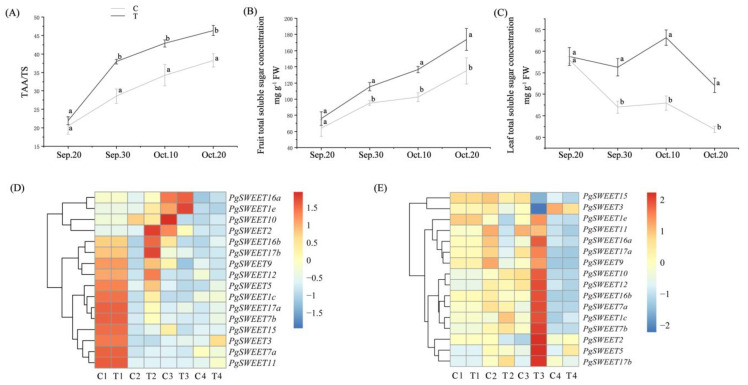
Changes in TSS/TA of fruit (**A**), total soluble sugar concentration of fruit (**B**), total soluble sugar concentration of leaf (**C**), SWEETs expression of fruit (**D**) and SWEETs expression of leaf (**E**) during fruit growth and development after pure water (C, control) and tripotassium phosphate (T, treatment). Each value is the mean for three replicates, with vertical bars representing standard errors. The different letters at each time point show significant differences (*p* < 0.05) using Duncan’s tests between control and treatment. In (**D**,**E**), numbers 1–4 represent dates from 20 September to 20 October 2020.

**Figure 5 ijms-23-02471-f005:**
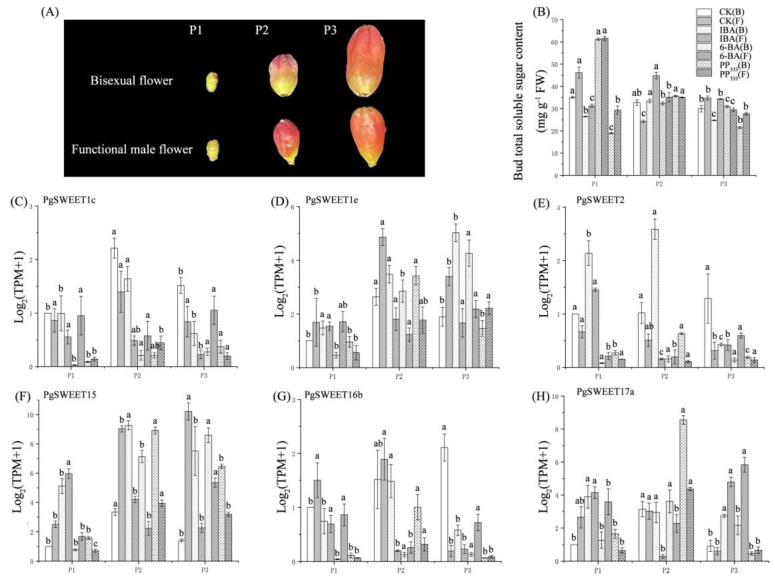
Buds of bisexual flower and functional male flowers during different stages (**A**). Changes in buds total soluble sugar content (**B**) and SWEETs expression (**C**–**H**) of bisexual and functional male flowers under hormone treatment. Each value is the mean for three replicates, with vertical bars representing standard errors. The different letters at each time point show significant differences (*p* < 0.05) using Duncan’s tests between P1, P2 and P3. In the note, B represents bisexual flower, F represents functional male flower.

**Figure 6 ijms-23-02471-f006:**
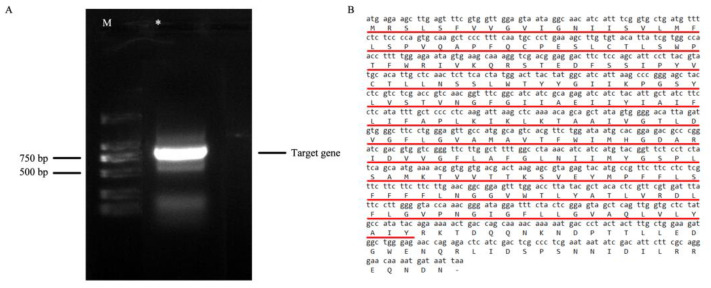
PCR amplification (**A**) and the coding sequence (**B**) of *PgSWEET17a.* Note: DL ladder 2000 DNA Marker; *: Pomegranate cDNA template; the red line is the MtN3/saliva domain.

**Figure 7 ijms-23-02471-f007:**
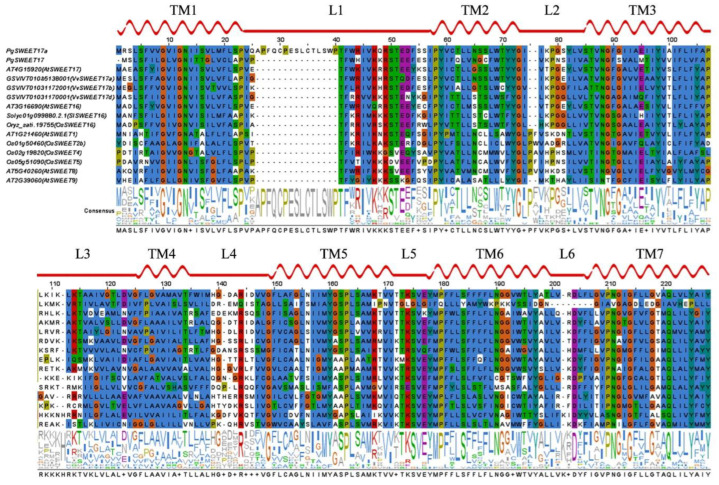
Sequence alignment of SWEET proteins.

**Figure 8 ijms-23-02471-f008:**
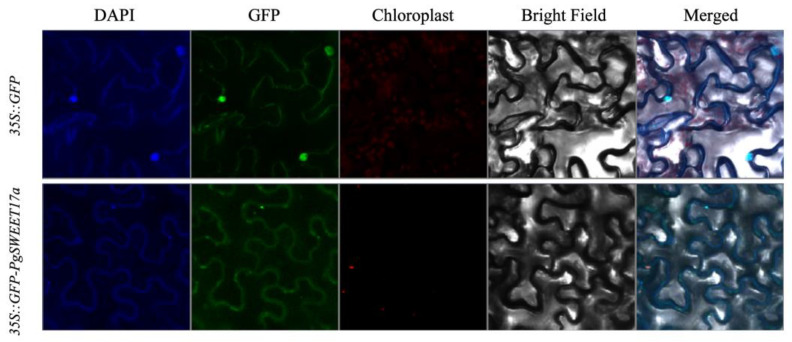
Subcellular localization of the 35S::GFP-PgSWEET17a fusion protein in tobacco leaves. Free GFP served as a control. A DAPI staining assay was conducted to confirm the nuclear localization.

**Figure 9 ijms-23-02471-f009:**
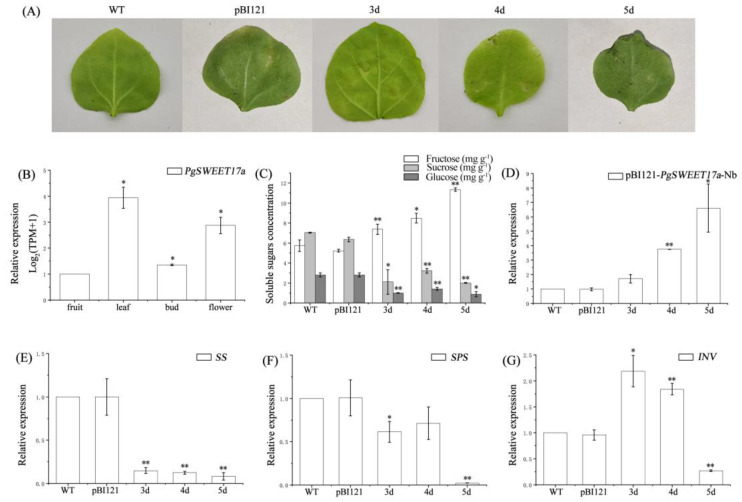
Tobacco transient expression results. Images of wild-type (WT), pBI121- overexpression and *PgSWEET17a*-overexpression tobacco leaves (**A**). *PgSWEET17a* expression of different organs (**B**). Changes in soluble sugar concentration (**C**), *PgSWEET17a* expression (**D**), *SS* expression (**E**), *SPS* (**F**) and *INV* (**G**) in WT, pBI121-overexpression and *PgSWEET17a*-overexpression tobacco leaves. “*” indicates significant differences at the 0.05 level, “**” indicate significant differences at the 0.01 level.

**Table 1 ijms-23-02471-t001:** SWEET genes involved in physiological function.

Clade	Gene	Involved in Sugar Transport	Reference	Gene	Involved in Reproductive Development	Reference
Ⅰ	*AtSWEET1*	plasma membrane	[8]	*AtSWEET1*	petal	
				*OsSWEET2a*	flower and panicle	[22]
Ⅱ				*ZmSWEET4c*	basal endosperm transfer layer	[23]
				*AtSWEET5*	mature pollen grain	[24]
				*LeSWEET5b*	mature pollen grain	[25]
				*OsSWEET5*	during flower and panicle development	[22]
				*AtSWEET7*	during pollen development	[26]
				*AtSWEET8*	during pollen wall and anther development	[27]
Ⅲ	*AtSWEET11*	phloem cell	[7]	*AtSWEET9*	nectary parenchyma cell	[15]
	*AtSEET12*	phloem cell	[7]	*NaSWEET9*	nectary cell	[15]
	*OsSWEET11*	phloem cell	[7]	*AtSWEET13*	stamen	[28]
	*OsSWEET14*	phloem cell	[7]	*AtSWEET14*	stamen	[16]
	*OsSWEET15*	senescent leaf	[21]	*OsSWEET11*	panicle and anther	[29]
Ⅳ	*AtSWEET16*	root	[30]			
	*AtSWEET17*	leaf and root	[20]			

Note: *At* represents *Arabidopsis*, *Os* represents rice, *Le* represents tomato, *Zm* represents maize.

**Table 2 ijms-23-02471-t002:** Basic information of the SWEET gene family in pomegranate.

Gene Name	Gene ID	Location	Exon No.	Protein Length/aa	Molecular Weight/ku	pI
*PgSWEET10*	*Pg000003.1*	scaffold1:6849801;6852242	6	318	35,423.91	8.46
*PgSWEET5*	*Pg000836.1*	scaffold1:5975742;5977070	6	240	27,038.52	9.73
*PgSWEET15*	*Pg002682.1*	scaffold11:951702;953916	6	304	34,149.26	7.65
*PgSWEET2*	*Pg006257.1*	scaffold14:1751672;1753913	6	235	26,053.98	9.12
*PgSWEET16a*	*Pg007971.1*	scaffold16:3103840;3105566	6	311	34,217.96	9.10
*PgSWEET16b*	*Pg007972.1*	scaffold16:3100513;3102712	6	314	34,865.04	9.29
*PgSWEET17b*	*Pg007973.1*	scaffold16:3089934;3093052	5	215	23,370.82	6.09
*PgSWEET1a*	*Pg008613.1*	scaffold165:311392;314797	5	253	28,013.80	9.33
*PgSWEET1d*	*Pg008635.1*	scaffold165:171819;177396	6	242	27,050.67	9.30
*PgSWEET1c*	*Pg008637.1*	scaffold165:213966;219285	6	259	28,898.66	9.25
*PgSWEET17a*	*Pg012205.1*	scaffold21:467086;471865	6	250	27,850.58	7.79
*PgSWEET8*	*Pg015006.1*	scaffold26:1968903;1972285	5	239	26,816.01	9.29
*PgSWEET1b*	*Pg015079.1*	scaffold260:29495;32112	5	227	25,323.43	9.18
*PgSWEET9*	*Pg017691.1*	scaffold33:1183976;1186154	6	270	30,135.24	9.52
*PgSWEET1e*	*Pg019984.1*	scaffold4:1483201;1490504	7	448	50,147.21	9.61
*PgSWEET3*	*Pg020538.1*	scaffold42:1971076;1972515	6	257	28,524.69	9.25
*PgSWEET12*	*Pg027298.1*	scaffold72:134285;136352	6	325	35,756.12	6.16
*PgSWEET7a*	*Pg029959.1*	scaffold9:1816650;1819681	3	213	23,424.94	9.25
*PgSWEET7b*	*Pg029960.1*	scaffold9:1824680;1828883	5	258	28,304.74	9.19
*PgSWEET11*	*Pg030654.1*	scaffold97:711886;713925	6	275	30,531.75	9.33

**Table 3 ijms-23-02471-t003:** RNA-seq data of pomegranate.

Accession No.	Cultivar	Sample	ID	Reference
SRR1054190	Black 127	Mixed samples of root, leaf, flower and fruit	Black 127	[43]
SRR1055290	nana	Mixed samples of root, leaf, flower and fruit	nana	[43]
SRR5279388	Dabenzi	Outer seed coat	DBZ_OSC	[42]
SRR5279391	Dbenzi	Inner seed coat	DBZ_ISC	[42]
SRR5279394	Dabenzi	Pericarp	DBZ_pericarp	[42]
SRR5279395	Dabenzi	Flower	DBZ_flower	[42]
SRR5279396	Dabenzi	Root	DBZ_root	[42]
SRR5279397	Dabenzi	Leaf	DBZ_leaf	[42]
SRR5446598	Tunisia	3–5 mm bud of bisexual flower	3–5 mm (B)	[44]
SRR5446595	Tunisia	5.1–13 mm bud of bisexual flower	5.1–13 mm (B)	[44]
SRR5446592	Tunisia	13.1–25 mm bud of bisexual flower	13.1–25 mm (B)	[44]
SRR5446607	Tunisia	3–5 mm bud of functional male flower	3–5 mm (F)	[44]
SRR5446604	Tunisia	5.1–13 mm bud of functional male flower	5.1–13 mm (F)	[44]
SRR5446601	Tunisia	13.1–25 mm bud of functional male flower	13.1–25 mm (F)	[44]

**Table 4 ijms-23-02471-t004:** Analysis of some important cis-acting regulatory elements in the promoter sequences of *PgSWEET17a*.

Cis-Acting Element Name	Sequence	Function
SREATMSD	TTATCC	Cis-acting element involved in the sugar signal
ABRE	ACGTG	Cis-acting element involved in the abscisic acid responsiveness
ARE	AAACCA	Cis-acting regulatory element essential for the anaerobic induction
AuxRR-core	GGTCCAT	Cis-acting regulatory element involved in auxin responsiveness
CGTCA-motif	CGTCA	Cis-acting regulatory element involved in the MeJA-responsiveness
G-Box	CACGTT	Cis-acting regulatory element involved in light responsiveness
LTR	CCGAAA	Cis-acting element involved in low-temperature responsiveness
MBS	CAACTG	MYB binding site involved in drought-inducibility

**Table 5 ijms-23-02471-t005:** Primers for the gene cloning, transient expression and qRT-PCR.

Primer	Primer Sequence(5′-3′)	Annotation
*PgSWEET17a*	F: ATGAGAAGCTTGAGT R: TTAATTATCATTTTGTTCCCT	Gene clone
*GFP-PgSWEET17a*	F: gagaacacgggggactctagaATGAGAAGCTTGAGT R: gcccttgctcaccatggatccTTAATTATCATTTTGTTCCCT	Transient expression
*qRT-PgSWEET17a*	F: GCCGTTCTTCCTCTCGTT R: TTTTGTTCCCTGCGATGGCT	Gene expression
*qRT-SS*	F:ATCAAGTTCCGGCCTTGGAG R:CCTCAGTGAATGTCTCCATG	Gene expression
*qRT-SPS*	F:GGAATTACAGCCCATACGAG R:AAGTTCTGGGTGAGCAAA	Gene expression
*qRT-INV*	F:CTCCACGACCCATTACAC R:GGAAACTCCCTGAGATACA	Gene expression
*PgActin*	F: AGTCCTCTTCCAGCCATCTC R: CACTGAGCACAATGTTTCCA	Gene expression
*NtActin*	F:CAAGGAAATCACCGCTTTGG R:AAGGGATGCGAGGATGGA	Gene expression

## Data Availability

The data presented in this study are available on request from the corresponding author and the public pomegranate transcriptomes presented in this study are available in insert article.

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
