# Peer review of "Identification, Analysis and Gene Cloning of the SWEET Gene Family Provide Insights into Sugar Transport in Pomegranate (Punica granatum)"

_ijms, 2022, doi:10.3390/ijms23052471_

Round 1
Reviewer 1 Report
Dear Editor and author,
Dear editor,
I have thoroughly reviewed this review manuscript titled “Identification, analysis and gene cloning of SWEET gene family provide insights into sugar transport in pomegranate (Punica granatum).
This article provides crucial insights into the sugar transport in pomegranate and identifies the SWEET gene family.
However, the manuscript has still some lacking that I noticed in the following comments;
Overall, I recommend this article as a “minor revision”.
Overall findings:
- The authors have done good work. The manuscript is well written
- The English quality seems great.
- The author should provide detailed information about the genes identified in this study as a supplementary file to justify the data.
Author Response
Dear reviewer,
RE: Manuscript ID ijms-1597357
We would like to thank you for giving us the opportunity to revise our manuscript. We have carefully taken your comments into consideration in preparing our revision. We have uploaded the word document for your review.

Reviewer 2 Report
This article reports the identification of the SWEET family in pomegranate using several bioinformatics methods. Furthermore, it describes the expression of some pgSWETT candidate genes, in tissues, cells and in different cultural conditions, in order to identify their role in the sugar distribution of plants. The "in vivo" experimental part seems a bit weak compared to the bioinformatics one.
Specific comments
Abstract
It is too long, some details could be shortened or omitted, for example " We determined the expression of PgSWEETs and found that PgSWEET2, 5, 9, 12, 16b, and 17b expressions in fruit were significantly up-regulated on September 30, and PgSWEET1c, 2, 5, 7a, 7b, 9, 11, 12, 16a, 16b, 17a, and 17b in leaf were significantly up-regulated on October 10".
The text often refers to tobacco plants, while in the materials and methods reference is made to the "Benthamiana", it seems that the plants used for the transformation are Nicotiana benthamiana, in this case use the scientific name N. benthamiana.
3.5 Changes in soluble sugar concentration and PgSWEETs expression pattern of pomegranate fruit and leaf after tripotassium phosphate treatment.
The method used to determine the sugar content is not described.
The method applied to determine the transcription levels of the 16 PgSWEET genes is not described.
As in: “.5. Plant material and tripotassium phosphate foliar fertilization "to better understand the text it seems useful to specify the phenological stage corresponding to the data i.e." 20 September 2020 (pomegranate fruit color change phase) ".
Figure 4 is difficult to read: it should be possible to change CK with C (control) and K3PO4 with T (treatment) using the same letter in all images (A - E), also in figures D and E it could be possible to replace the pgSWEET gene name (number) with number only.
3.6. Changes in buds total soluble sugar contents and PgSWEETs expression pattern…
The method used to determine the sugar content is not described.
The method applied to determine the expression of PgSWEET genes is not described.
The text and the figure 5 are difficult to read.
3.8. PgSWEET17a conservative domain analysis
Figure 7 is not legible
3.9. PgSWEET17a subcellular localization analysis
Results shown in Figure 8 are unclear: DAPI staining tests do not show nucleus, GFP localization appears to be the same in PgSWEET17a-GFP cells and pBI121-GFP control cells, merged images are not clear.
- Discussion
The sentence “PgSWEET2 was highly expressed in all tissues of flowers, buds, leaves and fruits, confirming its role in the reproductive development of pomegranate” is not supported by the data presented in Figure 4 D.
The expression pattern of the PgSWEETs family in the leaves and fruits of the pomegranate, after foliar application of tripotassium phosphate, reported in 3.6 seems to be quite random and the discussion does not explain these data well.
Author Response
Dear reviewer,
RE: Manuscript ID ijms-1597357
We would like to thank you for giving us the opportunity to revise our manuscript. We have carefully taken your comments into consideration in preparing our revision. We have uploaded the Word document for your review.
